# Effect of Ca Content on the Mechanical Properties and Corrosion Behaviors of Extruded Mg–7Li–3Al Alloys

**Xiaoming Xiong [1], Yan Yang [1,2,\*], Hongju Deng [1], Minmin Li [1], Jinguang Li [1], Guobing Wei [1,2] and Xiaodong Peng [1,2]**

[1]   International Joint Laboratory for Light Alloys (MOE), College of Materials Science and Engineering, Chongqing University, Chongqing 400044, China; 20160913127@cqu.edu.cn (X.X.); 201809131132@cqu.edu.cn (H.D.); 20180902041t@cqu.edu.cn (M.L.); 201709021013@cqu.edu.cn (J.L.); guobingwei@cqu.edu.cn (G.W.); pxd@cqu.edu.cn (X.P.)

[2]   National Engineering Research Center for Magnesium Alloys, Chongqing University, Chongqing 400044, China

\*   Correspondence: yanyang@cqu.edu.cn; Tel.: +86-23-6510-2856

**Abstract:** The effect of Ca addition on the microstructure, mechanical properties, and corrosion behaviors of the extruded Mg–7Li–3Al alloys was investigated. The results showed that the extruded Mg–7Li–3Al–xCa alloys consisted of $\alpha$-Mg (hcp) + $\beta$-Li (bcc) matrix phases and $Al_2Ca$. With increasing Ca content, the amount and morphology of the $Al_2Ca$ phase changed significantly. The grains of the extruded Mg–7Li–3Al–xCa alloys were refined by dynamic recrystallization during the extrusion process. The tensile tests results indicated that the extruded Mg–7Li–3Al–0.4Ca alloy exhibited favorable comprehensive mechanical properties; its ultimate tensile strength was 286 MPa, the yield strength was 249 MPa, and the elongation was 18.7%. The corrosion results showed that this alloy with 0.4 wt.% Ca addition exhibited superior corrosion resistance, with a corrosion potential $E_{corr}$ of $-1.48742$ $V_{VSE}$, attributed to the formation of protective $Al_2Ca$ phases.

**Keywords:** Mg–7Li–3Al alloy; extrusion; mechanical properties; corrosion

## 1. Introduction

Mg–Li alloys have been applied in automatic, 3C (Computer, Communication, Consumer Electronic) products and communication electronics fields for more than a decade owing to the high specific strength and advantageous formability [1–4]. However, their poor creep resistance and inferior corrosion resistance usually limit their application [5,6]. In order to solve the existing problems, many alloying elements (Al, Ca) have been used to improve the mechanical properties and corrosion resistance of Mg–Li alloys [7–9].

Some studies showed that Al can improve the mechanical properties of Mg–Li alloys owing to the formation of intermetallic compounds and a favorable solid-solution strengthening effect [10–14]. Bahman et al. [15] reported that more than 3 wt.% addition of Al dramatically degraded the corrosion resistance of Mg–Al–Zn–Ca alloys due to the formation of a $Mg_{17}Al_{12}$ network at the grain boundaries. Some research indicated that the formation of this $Mg_{17}Al_{12}$ phase can be inhibited by the addition of Ca [16–19].

It has been reported that Ca can improve the corrosion resistance of Mg alloys by the formation of protective $Al_2Ca$ and $(Mg, Al)_2Ca$ phases. At the same time, their high-temperature mechanical properties are also improved by grain refinement [20,21]. Li Zhang et al. [19] revealed that the microstructure and mechanical properties of Mg–Al–Ca alloys were affected by the Ca/Al ratio, indicating that the strengthening effect of the $Al_2Ca$ phase was better than that of the $Mg_2Ca$ phase. Wang et al. [22] showed that extruded Mg–5Li–3Al–0.5Ca alloy had the highest strength.

However, while reviewing the published literature, we found few reports about the influence of Ca and Al additions on the corrosion behaviors and mechanical properties of α (hcp) + β (bcc) dual-structured Mg–Li alloys. Thus, the effect of the addition of different Ca amounts on the mechanical properties and corrosion behavior of α (hcp) + β (bcc) dual-structured Mg-7Li-3Al-xCa (x = 0, 0.4, 0.8, 1.2 wt.%) alloys were investigated.

## 2. Experimental Procedures

Mg–7Li–3Al–xCa (x = 0, 0.4, 0.8, 1.2 wt.%) alloys [Mg, Al, and Li (99.9%) and Mg–30Ca master alloy (wt.%)] were melted in a melting furnace, protected by a $SF_6/CO_2$ gas mixture. The molten metal was cast in a cylinder metal mold and then was homogenized in a vacuum furnace at 250 °C for 4 h. Finally, the ingots, 80 mm in diameter, were extruded at 250 °C (extrusion ratio was 25:1). The chemical compositions of the alloys are listed in Table 1.

**Table 1.** Chemical composition of the extruded alloys.

| Materials | Chemical Composition (wt.%) | | | |
|---|---|---|---|---|
| | Li | Al | Ca | Mg |
| Mg–7Li–3Al | 6.85 | 2.78 | - | Bal. |
| Mg–7Li-3Al–0.4Ca | 7.21 | 2.85 | 0.38 | Bal. |
| Mg–7Li-3Al–0.8Ca | 6.49 | 3.12 | 0.87 | Bal. |
| Mg–7Li–3Al–1.2Ca | 6.76 | 2.75 | 1.15 | Bal. |

The microstructure of the alloys was analyzed by optical microscopy (OM; OLYMPUS PMG3, Shinjuku, Japan) and scanning electron microscopy (SEM; FEI NOVA 400, Tallahassee, FL, USA). Tensile tests at room temperature and high temperature (150 °C) were conducted on a tensile tester (SANS CMT-5105, Mohd Sultan Road, Singapore) with a displacement speed of 2 mm/min. In addition, the phase composition of the alloys was analyzed by X-ray diffraction (XRD; D/max-2500pc, Tokyo, Japan).

Cylindrical samples of Φ16 mm × 9 mm were machined from the extruded alloy bar and then ground to cylindrical samples of Φ15 mm × 7 mm. The size and weight of the samples were measured before the experiment, then the samples were placed in a 3.5 wt.% NaCl aqueous solution. After immersion for 72 h, the samples were cleaned with 200 g $L^{-1}$ $CrO_3$ and weighed again. The corrosion rate was calculated by:

$$c_w = 2.1 \times \Delta m / (At) \tag{1}$$

where $c_w$ is the corrosion rate in mm/year, $\Delta m$ is the weight loss in mg, $A$ is the surface area in $cm^2$, and $t$ is the immersion time in days. The electrochemical experiments were carried out by a GAMRY Reference 600+ system, and a three-electrode electrochemical cell was used at room temperature in a 3.5 wt % NaCl solution. Potentiodynamic polarization tests were implemented from −1.9 V vs. SCE (Saturated calomel electrode) to −1 V vs. SCE, with a scan rate of 0.5 mV/s. All electrochemical tests were performed 3 times to obtain the average of the corrosion potential ($E_{corr}$, V vs. SCE) and the corrosion current density ($i_{corr}$, A $cm^{-2}$). The frequency range for electrochemical impedance spectroscopy (EIS) testing was $10^5$–$10^{-2}$ Hz, and the amplitude of the applied sine wave disturbance was 10 mV. The measured data were analyzed, and the equivalent circuit parameters were obtained by Z-View software (Scribner Associates, Inc., Southern Pines, NC, USA).

## 3. Results

### 3.1. Microstructure

Optical micrographs of the extruded alloys are shown in Figures 1 and 2. According to the Figure 2a, the average grain size of the extruded Mg–7Li–3Al alloy was approximately 6.98 µm. After incrementing Ca, the average grain sizes of the Mg–7Li–3Al–0.4Ca and Mg–7Li–3Al–0.8Ca alloys were approximately 5.14 µm and 6.39 µm, respectively. The extruded Mg–7Li–3Al–1.2Ca alloy had the minimum grains size, with an average value of 3.92 µm, as shown in Figure 2d. According to previous research [23], the dark areas in the figures can be identified as α-Mg phase, and the grey areas as β-Li phase. It appeared that the α-Mg and β-Li phases were elongated along the extrusion direction, while the intermetallic compounds were mainly distributed at the boundaries of the α-Mg and β-Li phases and at the grain boundaries.

The XRD patterns of the extruded Mg–7Li–3Al–xCa alloys are presented in Figure 3. It is obvious that the Mg–7Li–3Al alloy mainly contained α-Mg, β-Li matrix, AlLi, and $Mg_{17}Al_{12}$ phases. With addition of 0.4 wt.% Ca, $Mg_{17}Al_{12}$ disappeared, while $Al_2Ca$ formed. When the amount of Ca was 0.8 wt.% and 1.2 wt.%, there was only $Al_2Ca$ as the second phase in the extruded alloys. The difference in electronegativity between Mg and Al is 0.3, while that between Al and Li is 0.63 [14]. Hence, the AlLi phase forms more easily than the $Mg_{17}Al_{12}$ phase. Thus, the compounds in the Mg–7Li–3Al alloy contained AlLi and a small amount of $Mg_{17}Al_{12}$. Similarly, the difference in electronegativity between Al and Ca is 0.5, while that between Mg and Ca is 0.21; thus, it is obvious that Al reacts with Ca more likely than Mg. Thus, $Al_2Ca$ forms more easily in alloys containing Ca [11].

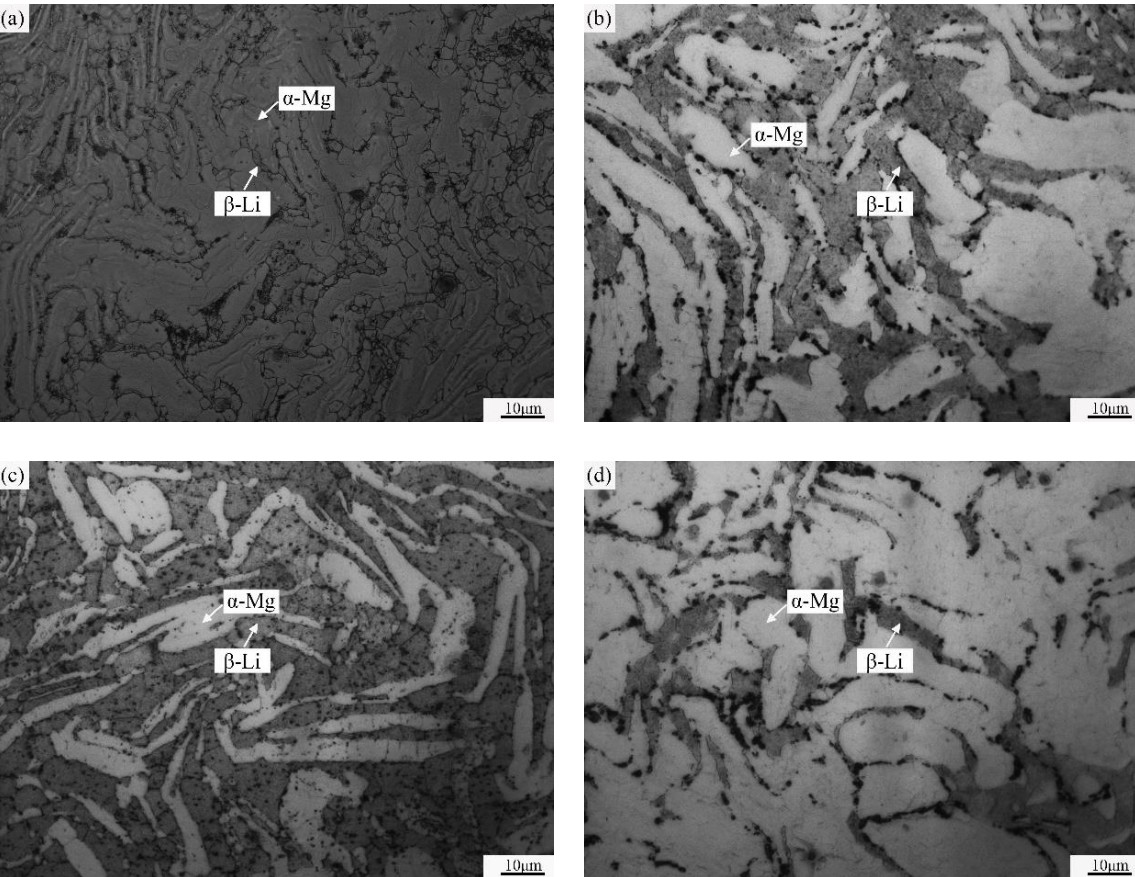

**Figure 1.** Microstructures of the extruded Mg–7Li–3Al–xCa alloys (perpendicular to the extrusion direction): (**a**) x = 0 (**b**) x = 0.4 (**c**) x = 0.8 (**d**) x = 1.2.

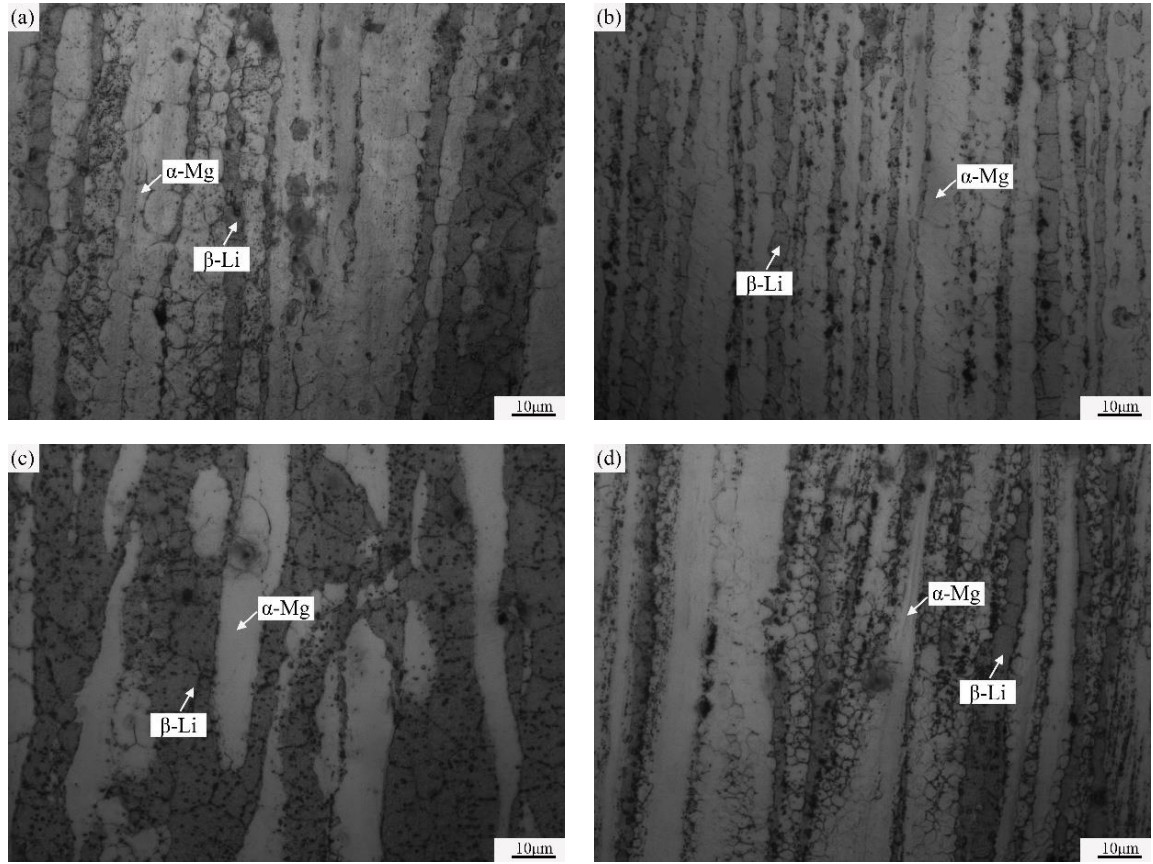

**Figure 2.** Microstructures of the extruded Mg–7Li–3Al–xCa alloys (parallel to the extrusion direction): (**a**) x = 0 (**b**) x = 0.4 (**c**) x = 0.8 (**d**) x = 1.2.

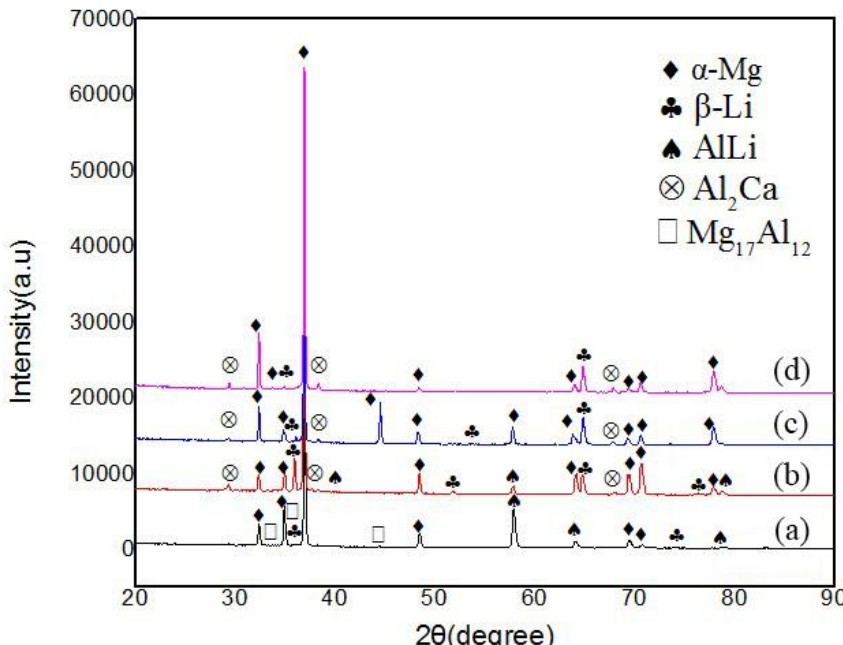

**Figure 3.** XRD patterns of the extruded Mg–7Li–3Al–xCa alloys: (**a**) x = 0 (**b**) x = 0.4 (**c**) x = 0.8 (**d**) x = 1.2.

The SEM images of the alloys are presented in Figure 4. The EDS analysis results of the points in Figure 4 are listed in Table 2. In the Mg–7Li–3Al alloy (Figure 4a), some bright particles (labeled as A)

were distributed at the phase boundaries and inside the α-Mg phase. The EDS results indicated that the compound contained Mg and Al at a mole ratio of ~17:12. Combined with the XRD data shown in Figure 3, the results indicate that the bright particles can be identified as $Mg_{17}Al_{12}$. In Figure 4b, the bright particles (marked as B) were identified as $Al_2Ca$ by combined EDS and XRD analyses. The EDS analysis showed that the short strip of intermetallic compound (marked as C) shown in Figure 4b consisted of Mg, Al, and Ca, with a mole ratio of Al to Ca of approximately 2.4:1, which indicated that this strip contained an $Al_2Ca$ phase. According to a previous study, [24] the mass ratio of Ca/Al in Mg–Al–Ca alloys determines the types of intermetallic compounds. Only $Al_2Ca$ exists in the alloys when the Ca/Al mass ratio is less than 0.8. Since the Ca/Al mass ratio in the Mg–7Li–3Al–0.4Ca alloy was approximately 0.13, the eutectic phase was expected to be $Al_2Ca$. Similarly, the compounds marked as D, E, and F in Figure 4c,d were inferred to contain $Al_2Ca$ by EDS and XRD analyses. With increasing Ca content, these $Al_2Ca$ compounds were observed to gradually aggregate into strips. The average size of these strip-shaped compounds in Figure 4d was approximately 16.31 μm.

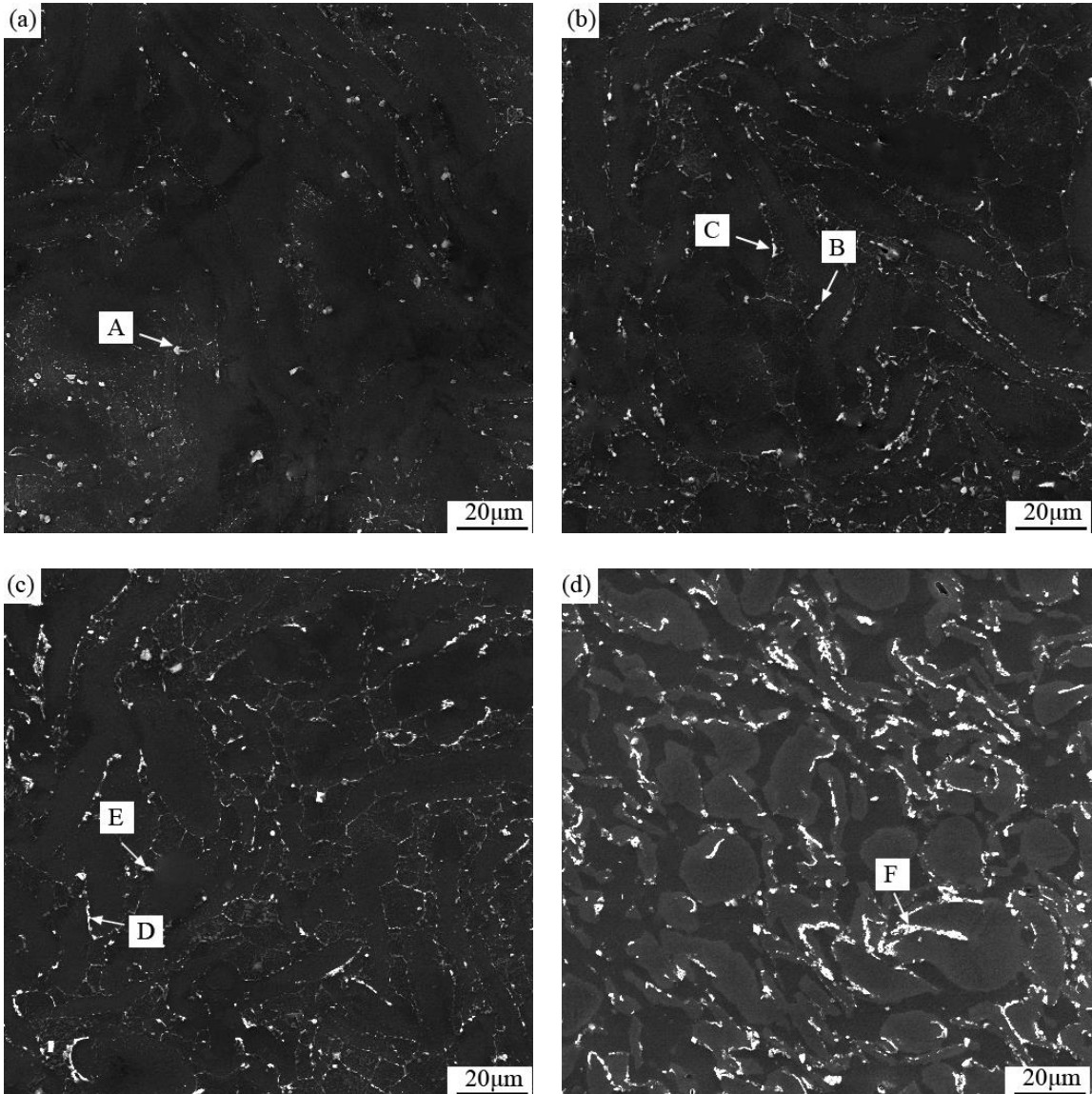

**Figure 4.** SEM images of the Mg–7Li–3Al–xCa alloys: (**a**) x = 0 (**b**) x = 0.4 (**c**) x = 0.8 (**d**) x = 1.2.

**Table 2.** EDS result corresponding to different positions in Figure 4. (mole fraction, %).

| Position | Mg | Al | Ca |
|----------|-------|-------|-------|
| A | 61.12 | 38.88 | 0 |
| B | 0 | 65.88 | 34.12 |
| C | 75.44 | 16.67 | 7.89 |
| D | 59.51 | 27.38 | 13.11 |
| E | 0 | 51.46 | 48.54 |
| F | 0 | 68.54 | 36.46 |

### 3.2. Mechanical Properties

Figure 5 presents the stress–strain curves of the extruded alloys. The values obtained from these curves are listed in Table 3. When the amount of Ca was 0.4%, the ultimate tensile strength (UTS) of the alloy reached a maximum of 286 MPa, while the UTS of the alloy without Ca was 268 MPa. Therefore, the addition of 0.4% of Ca can increase the tensile strength of the Mg–7Li–3Al alloy. The UTS and elongation of the extruded Mg–7Li–3Al–0.4Ca alloy were improved by ~6.8% and 14.7%, respectively, compared to those of the Mg–7Li–3Al alloy. When the addition of Ca exceeded 0.8%, the tensile yield strength (TYS) and the UTS exhibited a slight decline. The TYS and UTS of the Mg–7Li–3Al–1.2Ca alloy decreased 26 MPa and 30 MPa, respectively, compared to those of the Mg–7Li–3Al alloy, while the elongation improved by addition of Ca.

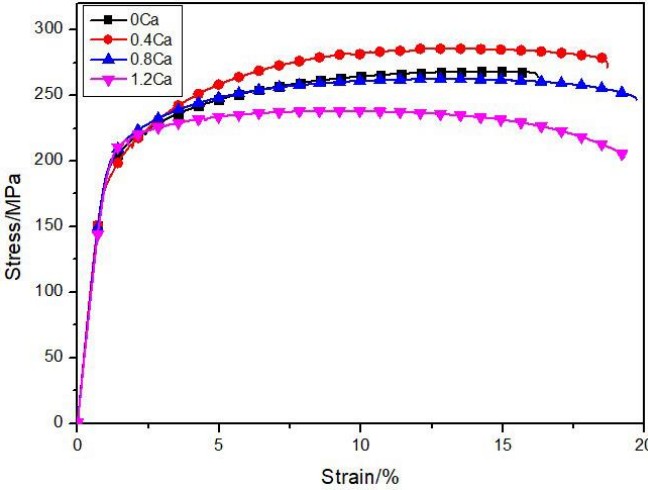

**Figure 5.** Stress–strain curves of the extruded Mg–7Li–3Al–xCa alloys at room temperature.

**Table 3.** Mechanical properties of the extruded Mg–7Li–3Al–xCa test alloys at room temperature.

| Extruded Alloys | TYS (MPa) | UTS (MPa) | Elongation (%) |
|-----------------|-----------|-----------|----------------|
| Mg–7Li–3Al | 233 | 268 | 16.3 |
| Mg–7Li–3Al–0.4Ca | 249 | 286 | 18.7 |
| Mg–7Li–3Al–0.8Ca | 229 | 263 | 19.7 |
| Mg–7Li–3Al–1.2Ca | 207 | 238 | 19.3 |

Figure 5 presents the stress–strain curves of the extruded alloys at 423 K (150 °C). The Mechanical properties of the extruded Mg–7Li–3Al–xCa alloys at 423 K (150 °C) are listed in Table 4. According to Figure 6 and Table 4, with increasing Ca content, the TYS of the experimental alloys at 423 K (150 °C) first increased and then decreased. The extruded Mg–7Li–3Al–0.4Ca alloy exhibited good high-temperature mechanical properties, with a TYS of 191 MPa, which improved by 22.4% compared to the TYS of the Mg–6.8Li–3Al alloy. The $Al_2Ca$ phase, with a melting point of 1079 °C and notable high-temperature stability, promotes strength improvement. The elongation of the test alloys first

decreased and then increased with increasing Ca content. The extruded alloy with a Ca content of 1.2 wt.% underwent the largest elongation, reaching a value of 24.8%.

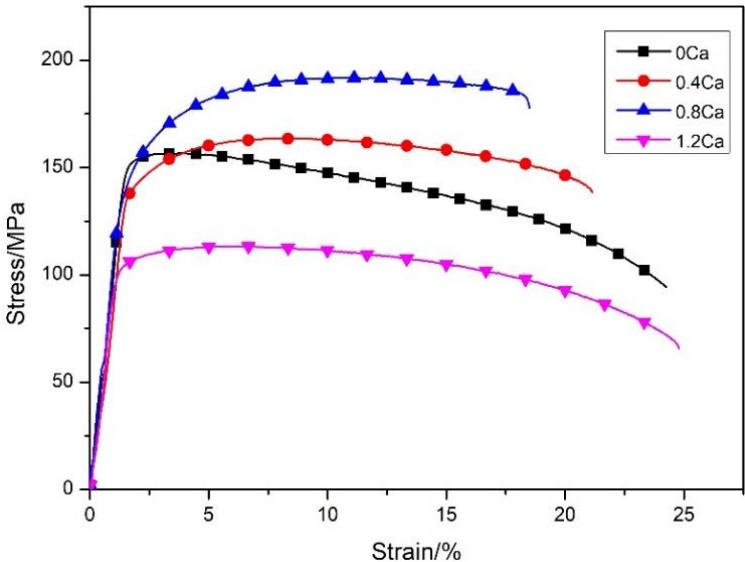

**Figure 6.** Stress–strain curves of extruded Mg–7Li–3Al–xCa alloys at 423 K (150 °C).

**Table 4.** Mechanical properties of the extruded Mg–7Li–3Al–xCa alloys at 423 K (150 °C).

| Extruded Alloys | TYS (MPa) | UTS (MPa) | Elongation (%) |
|---|---|---|---|
| Mg–7Li–3Al | 145 | 156 | 24.3 |
| Mg–7Li–3Al–0.4Ca | 141 | 163 | 21.1 |
| Mg–7Li–3Al–0.8Ca | 163 | 191 | 18.5 |
| Mg–7Li–3Al–1.2Ca | 98 | 113 | 24.8 |

*3.3. Corrosion Behavior*

Figure 7 shows the potentiodynamic polarization behaviors of the test alloys in a 3.5 wt.% NaCl solution. The values obtained from these curves are listed in Table 5. The corrosion potential ($E_{corr}$, V vs. SCE) for the Mg–7Li–3Al alloy was −1.5445 $V_{VSE}$, while those for the Mg–7Li–3Al–0.4Ca, Mg–7Li–3Al–0.8Ca, and Mg–7Li–3Al–1.2Ca alloys containing Ca, were −1.48742 $V_{VSE}$, −1.4940 $V_{VSE}$, and −1.4799 $V_{VSE}$, respectively. The $E_{corr}$ of the alloys improved substantially by the addition of Ca. Furthermore, with increasing Ca content, the corrosion current density ($i_{corr}$, A cm$^{-2}$) of the alloys decreased. The breakdown potential ($E_b$) was detected at −1.3735 V (vs. SCE) for the extruded Mg–7Li–3Al–1.2Ca alloy, as shown in Table 5. The extruded alloy with a Ca content of 1.2 wt.% and the lowest corrosion current density (31 μA cm$^2$) exhibited the best corrosion resistance among the alloys. It can be inferred that the passive film formed on the Mg–7Li–3Al–1.2Ca alloy was relatively stable due to its high $E_b$ value.

**Table 5.** Values obtained from the potentiodynamic polarization tests (in Figure 7).

| Alloys | $E_{corr}$ (V vs. SCE) | $E_b$ (V vs. SCE) | $\beta_c$ (mV/dec) | $i_{corr}$ (μA/cm$^2$) |
|---|---|---|---|---|
| Mg–7Li–3Al | −1.5445 | - | −342.18 | 134.90 |
| Mg–7Li–3Al–0.4Ca | −1.48742 | - | −240.84 | 39.81 |
| Mg–7Li–3Al–0.8Ca | −1.4940 | - | −294.18 | 63.10 |
| Mg–7Li–3Al-1.2Ca | −1.4799 | −1.3735 | −260.61 | 31.62 |

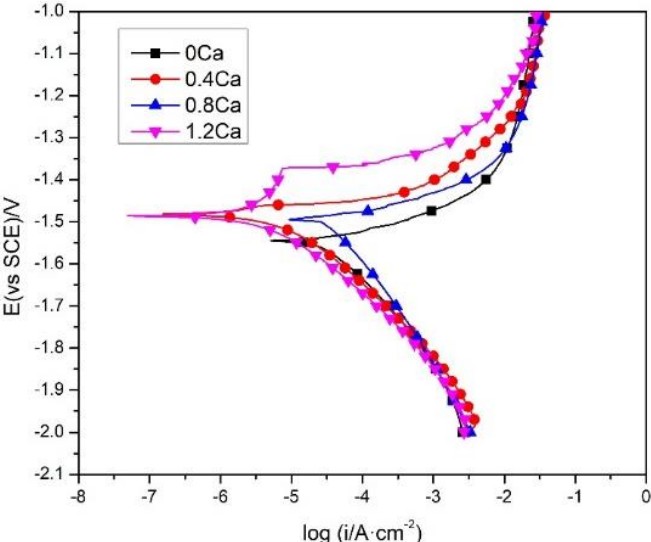

**Figure 7.** Potentiodynamic polarization curves of the extruded Mg–7Li–3Al–xCa alloys. The measurements were performed in a 3.5 wt.% NaCl solution.

The Nyquist and Bode plots of the samples are presented in Figure 8. As shown in Figure 8a, the Nyquist plots of all the experimental alloys consist of one capacitive loop in the high-frequency region and another capacitive loop in the low-frequency region. The appearance of the capacitive loop in the high-frequency region is mainly due to the transfer of charge, while its presence in the low-frequency region is mainly due to material transfer; the low-frequency resistance reflects the initiation of corrosion. The radius value of the loops of the experimental alloys containing Ca were larger than those of the Mg–7Li–3Al alloy, indicating that the corrosion resistance of the Mg–7Li–3Al alloys containing Ca was better than the corrosion resistance of the Mg–7Li–3Al alloy. In addition, the largest radius value corresponded to the Mg–7Li–3Al–1.2Ca alloy, and the corrosion resistance of the alloys was as follows: Mg–7Li–3Al–1.2Ca > Mg–7Li–3Al–0.4Ca > Mg–7Li–3Al–0.8Ca > Mg–7Li–3Al. These results are consistent result with the polarization curves.

From Figure 8b, the impedance value of the Mg–7Li–3Al–1.2Ca alloy was the highest at low frequency, among all samples. In general, the larger the impedance value in the Bode plot, the better the densification of the passivation film of the alloy. In Figure 8c, a prominent wave crest can be seen for all the samples at high frequencies because of the double-layer capacitance and the corresponding charge transfer resistance during the generation process of the corrosion products. Additionally, when the alloy contained Ca, a small peak appeared in the low-frequency range, which corresponded to an increase in film resistance due to the accumulation of corrosion products.

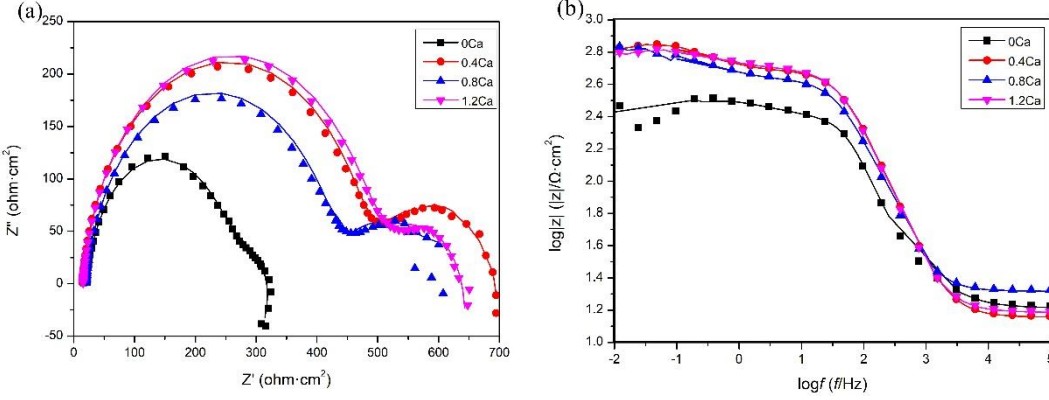

**Figure 8.** *Cont.*

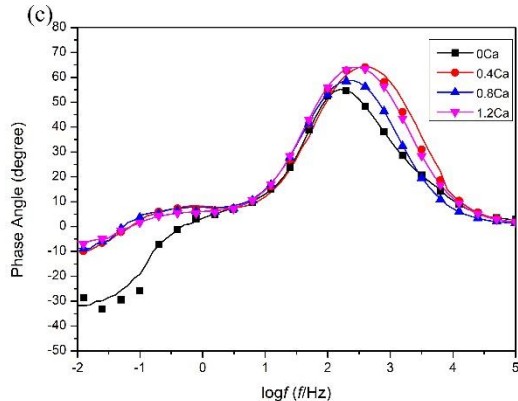

**Figure 8.** Impedance plots of the extruded Mg–7Li–3Al–xCa alloys immersed in a 3.5 wt.% NaCl solution. (**a**) Nyquist plots, (**b**) Bode plots, (**c**) Phase angle diagram.

Figure 9 shows the EIS spectra of the alloys, and the fitted results of the EIS measurement of the extruded Mg–7Li–3Al–xCa alloys are listed in Table 6. Here, $R_s$, $R_{ct}$, $R_f$, and *CPE* represent the solution resistance, charge transfer resistance, film resistance, and electric double-layer capacity, respectively. In general, higher values of $R_{ct}$ and $R_f$ indicate better corrosion resistance, while lower values of $CPE_1$ and $CPE_2$ indicate a Please check if the original meaning is retained and thicker film on the surface of the alloys [25,26]. The Mg–7Li–3Al–0.4Ca, Mg–7Li–3Al–0.8Ca, and Mg–7Li–3Al–1.2Ca alloys exhibited higher $R_{ct}$ and $R_f$ values than the Mg–7Li–3Al alloy, implying that the corrosion resistance of the extruded Mg–7Li–3Al alloy can be increased by the addition of Ca. Furthermore, the largest $R_{ct}$ and $R_f$ values were found for the Mg–7Li–3Al–1.2Ca alloy, demonstrating that the addition of 1.2 wt.% Ca significantly improved the corrosion resistance of the alloy.

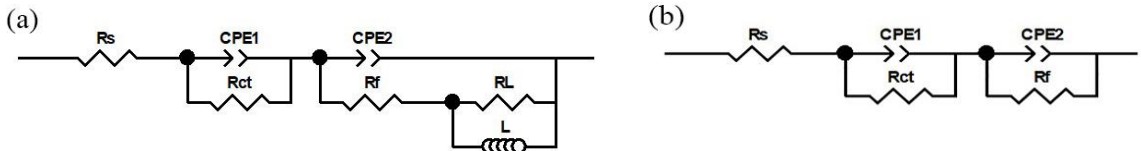

**Figure 9.** Equivalent circuits of the EIS spectra of (**a**) Mg–7Li–3Al, Mg–7Li–3Al–0.4Ca, and Mg–7Li–3Al–1.2Ca alloys, and (**b**) Mg–7Li–3Al–0.8Ca alloy.

**Table 6.** Fitted results of the EIS measurement of the extruded Mg–7Li–3Al–xCa alloys using the equivalent circuits shown in Figure 9.

| Ca Content | $R_S$ ($\Omega$) | $R_{ct}$ ($\Omega$ cm$^2$) | $CPE_1$ ($10^{-6}$ s$^n$ $\Omega^{-1}$ cm$^{-2}$) | $n_1$ | $R_f$ ($\Omega$ cm$^2$) | $CPE_2$ (s$^n$ $\Omega^{-1}$ cm$^{-2}$) | $n_2$ | $L$ |
|---|---|---|---|---|---|---|---|---|
| 0Ca | 17.09 | 257.7 | 26.556 | 0.87955 | 14.6 | 0.00614 | 0.60395 | 20.75 |
| 0.4Ca | 14.06 | 447.6 | 10.306 | 0.94147 | 235 | 0.00427 | 0.53179 | 231.9 |
| 0.8Ca | 20.65 | 401.7 | 14.27 | 0.92142 | 189.1 | 0.00472 | 0.66559 | - |
| 1.2Ca | 15.32 | 475.1 | 11.27 | 0.93109 | 241.4 | 0.00649 | 0.60237 | 140.6 |

*3.4. Immersion Test*

As shown in Figure 10a, the hydrogen evolution of the Mg–7Li–3Al alloy was reached higher values than that of the alloys containing Ca. It can be seen from Figure 10a that in the first 12 h, the hydrogen evolution rate was relatively slow, while between 12 and 30 h, the hydrogen evolution rate significantly increased, because of the strong corrosiveness and high conductivity of Cl$^-$, which accelerated ion transport and diffusion of Mg$^{2+}$ during the corrosion process and promoted the corrosion of the alloy. With the extension of time, the hydrogen evolution rate after 30 h decreased significantly owing to the increase of corrosion products formed on the surface of the alloy that filled the gaps on the surface oxide film to some extent, slowing down the corrosion process. In addition, the corresponding corrosion rates

were 5.50, 1.73, 2.93, and 1.56 mm/year for the Mg–7Li–3Al, Mg–7Li–3Al–0.4Ca, Mg–7Li–3Al–0.8Ca, and Mg–7Li–3Al–1.2Ca alloys, respectively. The $C_w$ rate and the volume of hydrogen evolution for the alloys containing Ca were slightly lower than those of the Mg–7Li–3Al alloy, indicating that Ca addition had a significant effect on the corrosion resistance.

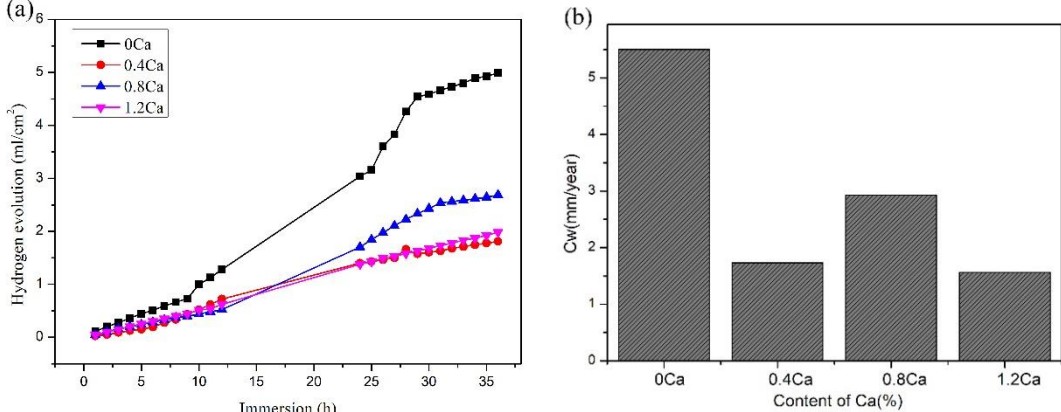

**Figure 10.** (**a**) Hydrogen evolution of the extruded Mg–7Li–3Al–xCa alloys immersed in a 3.5 wt.% NaCl solution, (**b**) Corrosion rates ($C_w$) of the extruded Mg–7Li–3Al–xCa alloys immersed in a 3.5 wt.% NaCl solution for 7 days.

## 4. Discussion

### 4.1. Effect of Ca Content on the Strength of Extruded Mg–7Li–3Al–xCa Alloys

The addition of Ca refines the microstructure of Mg–7Li–3Al-based alloys. In addition, dynamic recrystallization leads to grain refinement during hot extrusion, as indicated by the Hall–Petch formula:

$$\sigma = \sigma_0 + Kd^{-1/2} \tag{2}$$

where $\sigma$ is the yield stress, $\sigma_0$ is the yield strength of a single crystal ($\sigma_0$ = 11 MPa/$\mu$m$^2$ for Mg), $K$ is the Hall–Petch slope ($K$ = 0.28 MNm$^{-3/2}$ for Mg [27]), and $d$ is the average grain size. Therefore, this is probably the reason why the Mg–7Li–3Al–0.4Ca alloy showed the highest strength among the alloys.

With the addition of Ca, the Al$_2$Ca phase formed. When this second phase (Al$_2$Ca phase) was uniformly distributed in the matrix phase with finely dispersed particles, a significant strengthening effect was produced. The strength of the alloys can be estimated from the following Equation [5]:

$$\sigma = f_m\sigma_m + f_h\sigma_h \tag{3}$$

where $\sigma$, $\sigma_m$, and $\sigma_h$ are the strength of the Mg–7Li–3Al–xCa alloys, Mg–7Li–3Al matrix alloy, and Al$_2$Ca phase, respectively, and $f_m$ and $f_h$ are the volume fractions of the matrix alloys and Al$_2$Ca particulates, respectively. The volume fraction of Al$_2$Ca increased with increasing Ca content, resulting in an improvement in the strength of the alloys.

However, the yield strength of the Mg–7Li–3Al–1.2Ca alloy was lower than those of the Mg–7Li–3Al and Mg–7Li–3Al–0.4Ca alloys at room temperature, which contrasts with the results discussed above. According to a previous study [28], the morphology and size of the second phase have a great influence on the mechanical properties of an alloy. Large second-phase particles in the alloy can decrease the strength of the material. In particular, the high concentration of stress that may occur on these large particles during plastic tensile deformation ultimately leads to crack initiation and fracture at relatively early stages of the tensile test. The average size of these strip-shaped compounds shown in Figure 4d was approximately 16.31 $\mu$m, which resulted in worse mechanical properties of the Mg–7Li–3Al–1.2Ca alloy compared to the Mg–7Li–3Al and Mg–7Li–3Al–0.4Ca alloys.

### 4.2. Corrosion Mechanism of the Extruded Mg–7Li–3Al–xCa Alloys

According to Figure 11, the corrosion area of the extruded Mg–7Li–3Al alloy was larger than that of the other alloys, which implies that the corrosion resistance of the extruded Mg–7Li–3Al alloy was the worst. For the Mg–7Li–3Al–0.4Ca alloys, many corrosion pits and some amount of $Al_2Ca$ existed in the matrix, as presented in Figure 11b, demonstrating that the matrix corroded due to its lower electrical potential with respect to that of $Al_2Ca$. In Figure 11c, the corroded surface of the Mg–7Li–3Al–0.8Ca alloy was similar to that of the Mg–7Li–3Al–0.4Ca alloy. On the corroded surface of the Mg–7Li–3Al–1.2Ca alloy, the white areas corresponding to $Al_2Ca$ significantly increased and gathered into a line shape, which led to the aggregation of corrosion pits.

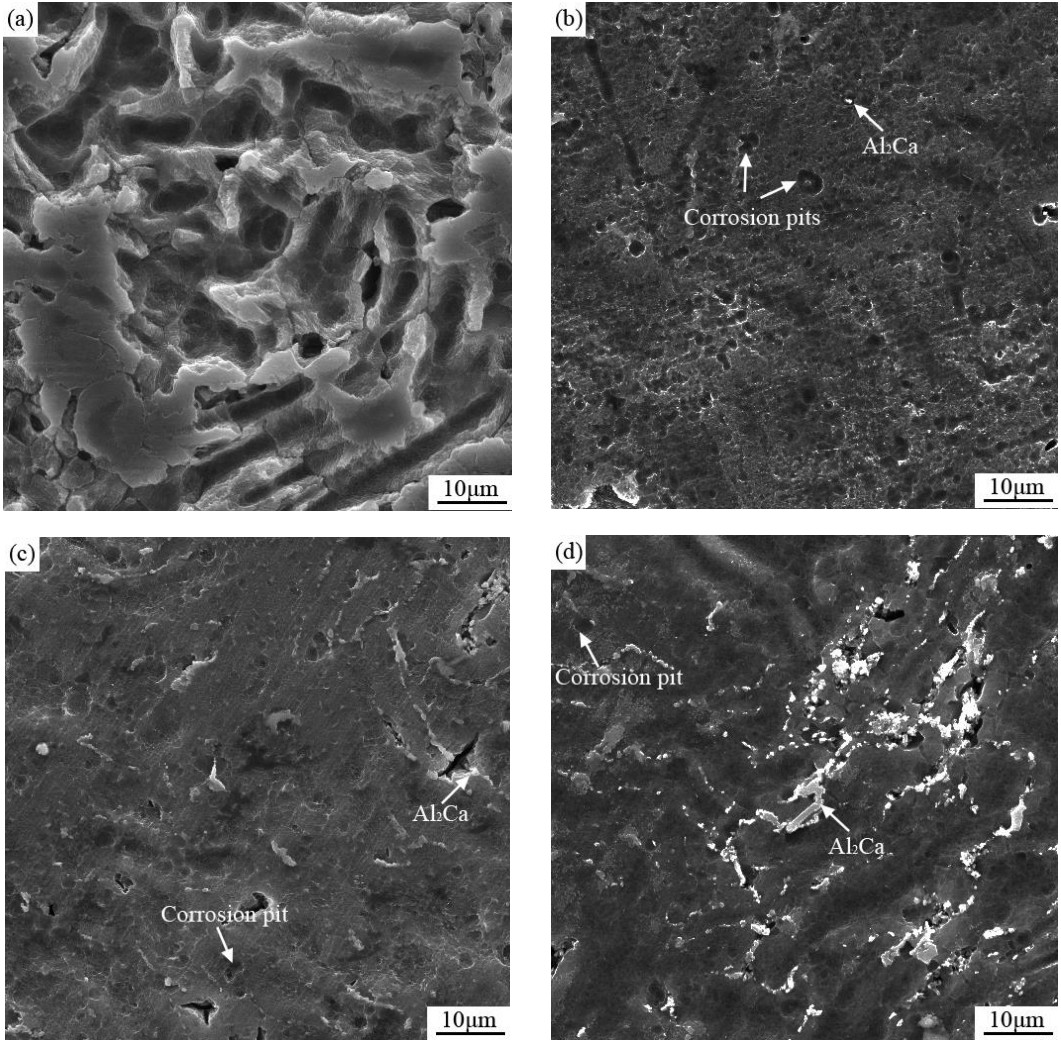

**Figure 11.** Corroded surfaces of the extruded Mg–7Li–3Al–xCa alloys after immersion for 72 h in a 3.5 wt.% NaCl solution at 25 °C: (**a**) x = 0 (**b**) x = 0.4 (**c**) x = 0.8 (**d**) x = 1.2.

The corrosion of the extruded Mg alloys is a dissolution process, mainly occurring on the bare parts of the Mg substrate. The reaction equations are as follows [16,29]:

$$\text{Cathodic reaction}: \ 2H_2O + 2e^- \rightarrow H_2 \uparrow + 2OH^- \tag{4}$$

$$\text{Anodic reaction}: \ Mg \rightarrow Mg^{2+} + 2e^- \tag{5}$$

$$\text{Corrosion product formation}: \ Mg + 2OH^- \rightarrow Mg(OH)_2 \downarrow \tag{6}$$

As for the extruded Mg–7Li–3Al alloy, additional reactions (7)–(10) can occur.

$$\text{Anodic reaction}: \ \text{Li} \rightarrow \text{Li}^+ + \text{e}^- \tag{7}$$

$$\text{Al} \rightarrow \text{Al}^{3+} + 3\text{e}^- \tag{8}$$

$$\text{Corrosion product formation}: \ \text{Li} + \text{OH}^- \rightarrow \text{Li(OH)} \downarrow \tag{9}$$

$$\text{Al} + 3\text{OH}^- \rightarrow \text{Al(OH)}_3 \downarrow \tag{10}$$

Generally, these corrosion products can form a protective film that prevents corrosion in the presence of a corrosive solution. However, the main corrosion product, $Mg(OH)_2$, has many pores, enabling the surrounding solution to continuously penetrate the surface film and react with the exposed inner matrix.

When Ca was added to the Mg–7Li–3Al alloy, the formed $Al_2Ca$ phase had a significant effect on the corrosion of the extruded alloy. Taking the Mg–7Li–3Al–0.4Ca alloy as an example, the corrosion mechanism of Ca-containing extruded alloys is illustrated in Figure 12. Figure 13 presents micrographs of the corrosion at different stages for the extruded Mg–7Li–3Al–0.4Ca alloy after immersion in the 3.5 wt.% NaCl solution. In general, intermetallic phases, such as $Al_2Ca$ particles, are chemically more stable than the α-Mg + β-Li matrix. Notably, previous studies have indicated that the corrosion resistance of the α-Mg phase is superior to that of the β-Li phase [29,30]. Thus, in stage 1, as shown in Figure 12, micro-galvanic couples first formed between the $Al_2Ca$ particles and the β-Li phases in the matrix. Corrosion first appeared at the surface of the sample, as presented in Figure 13a. Furthermore, in stage 2, the β-Li phase containing $Al_2Ca$ particles corroded and dissolved, leading to the formation of corrosion pits, as shown in Figure 13b. In addition, micro-galvanic couples started to form between the $Al_2Ca$ particles and the α-Mg phase. In stage 3, the α-Mg phase containing $Al_2Ca$ particles dissolved and formed corrosion pits. Moreover, the corrosion rate of the β-Li phase increased significantly, resulting in the formation of grooves (Figure 13c) and in $Al_2Ca$ particle detachment. After the corrosion of the matrix containing $Al_2Ca$ particles, the interface of the β-Li and α-Mg phases was susceptible to localized corrosion. This localized corrosion initiated at the boundary of the α-Mg and β-Li phases and then extended toward the more active β-Li phase, as shown in Figure 13d,e. In the stage 4, after the α-Mg + β-Li phases dissolved completely, $Al_2Ca$ separated from the matrix; a post-corrosion micrograph is presented in Figure 13f. Compared with Mg–7Li–3Al–0.4Ca, Mg–7Li–3Al–0.8Ca contains a high amount of secondary phases and more micro-galvanic couples, which determined a higher corrosion rate, while the Mg–7Li–3Al–0.4Ca alloy showed the best corrosion resistance. The reason for this phenomenon can be that the addition of 1.2Ca led to grain refinement. Previous reports demonstrated that the grain boundaries can not only provide nucleation sites for the passivation films [31] but also act as a physical corrosion barrier [32], indicating that a small grain size can improve the corrosion resistance.

Compared with the corrosion behavior of the extruded alloys containing Ca, the dissolution of the α-Mg + β-Li matrix in the extruded Mg–7Li–3Al alloy resulted in a larger corrosion area. When Ca was added to the alloys, the corrosion resistance of the extruded alloys significantly improved. This occurred because the corrosion mechanism changed from local corrosion, initiated at the phase boundaries of the extruded Mg–7Li–3Al alloy, to pitting corrosion starting on the $Al_2Ca$ particles of the extruded alloys containing Ca. Therefore, the corrosion resistance of the extruded alloys containing Ca was better than that of the extruded Mg–7Li–3Al alloy.

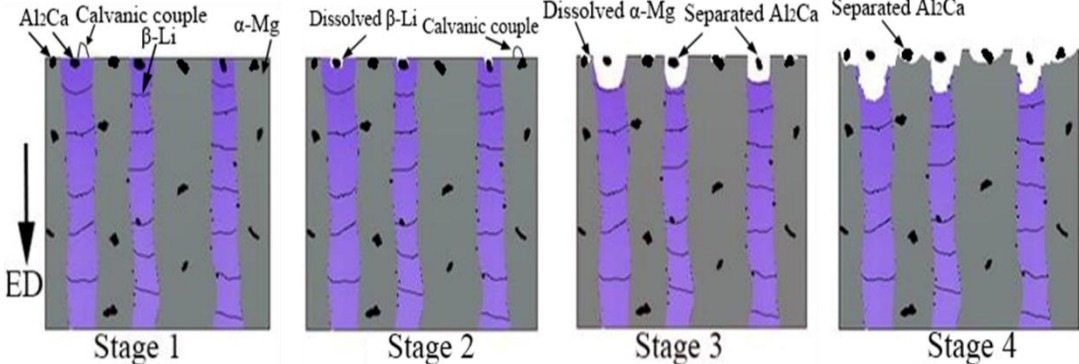

**Figure 12.** Schematic illustration of the role of the secondary phases in the extruded Mg–7Li–3Al–xCa alloys where their distribution is discontinuous.

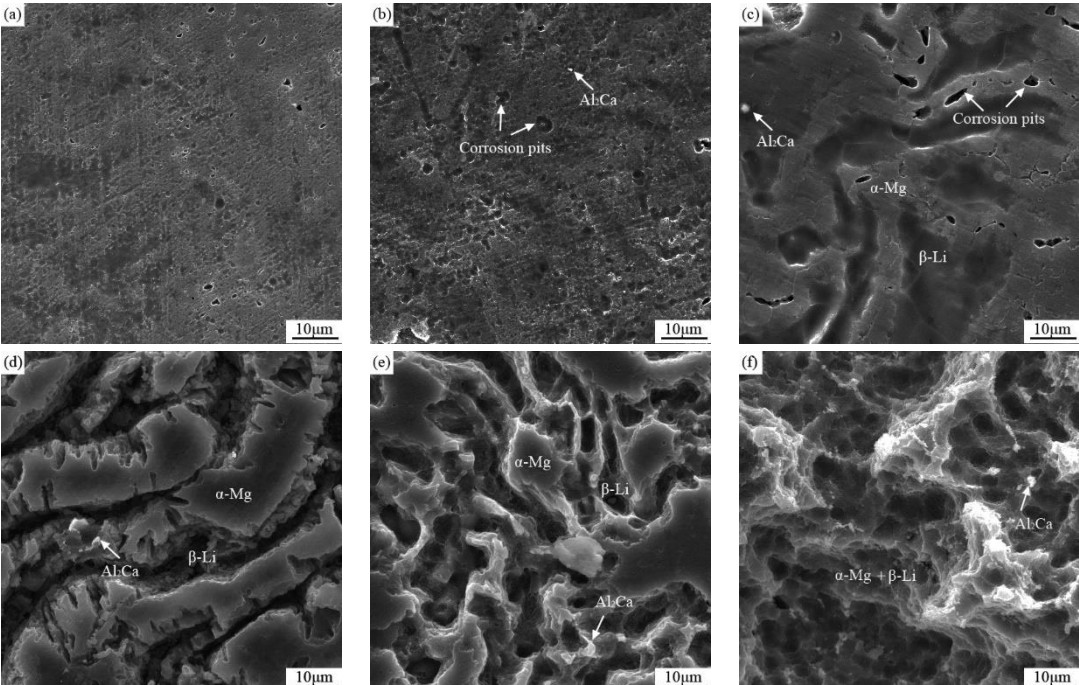

**Figure 13.** Micrographs showing corrosion at different times in the extruded Mg–7Li–3A–0.4Ca alloy after immersion in a 3.5 wt.% NaCl solution at 25 °C. (**a**) 6 h (**b**) 12 h (**c**) 24 h (**d**) 48 h (**e**) 60 h (**f**) 72 h.

## 5. Conclusions

(1) The grains of extruded Mg–7Li–3Al–xCa alloys were significantly refined as a result of dynamic recrystallization during the extrusion process. The $\alpha$-Mg and $\beta$-Li phases were elongated along the extrusion direction. $Al_2Ca$ in the alloys formed and mainly distributed at the boundaries of the $\alpha$-Mg and $\beta$-Li phases and at the grain boundaries.

(2) With increasing Ca content, the strength of the extruded Mg–7Li–3Al–xCa alloys first increased and then decreased. The extruded Mg–7Li–3Al–0.4Ca alloy exhibited favorable mechanical performance, demonstrating a UTS of 286 MPa, a TYS of 249 MPa, and elongation of 18.7%. The extruded Mg–7Li–3Al–0.8Ca alloy exhibited favorable mechanical properties at 423 K (150 °C), with an UTS of 191 MPa.

(3) The addition of Ca can improve the corrosion resistance of the extruded Mg–7Li–3Al alloy, which is attributed to the formation of $Al_2Ca$ particles. The corrosion mechanism of the extruded Mg–7Li–3Al alloy is local corrosion initiated at the phase boundaries, while for the extruded alloys containing Ca, the corrosion mechanism is pitting corrosion starting on the $Al_2Ca$ particles.

**Author Contributions:** Y.Y., X.P. and X.X. conceived and designed the experiments; X.X. and H.D. performed the experiments; X.X., H.D., M.L. and J.L. analyzed the data; X.X. and G.W. contributed reagents/materials/analysis tools; X.X. wrote the paper, Y.Y., X.P. and H.D. writing-review and editing the paper.

**Funding:** National Natural Science Foundation of China: No. 51601024; National Key Research and Development Program of China: No. 2016YFB0700403; National Key Research and Development Program of China: No. 2016YFB0301100; Fundamental Research Funds for the Central Universities of china: No. 2018CDJDCL0020; Fundamental Research Funds for the Central Universities of china: No. 2018CDJDCL0019; Fundamental Research Funds for the Central Universities of china: No. 2018CDGFCL0005; Ministry of Education and the State Administration of Foreign Experts Affairs of China: No. B16007.

**Acknowledgments:** The authors acknowledge financial support by the National Natural Science Foundation (Project No. 51601024), the National Key Research and Development Program of China (Project No. 2016YFB0700403 & Project No. 2016YFB0301100), the Fundamental Research Funds for the Central Universities (Project No. 2018CDJDCL0019, Project No. 2018CDJDCL0020 and Project No. 2018CDGFCL0005) and the support of the 111 Project (Project No. B16007) by the Ministry of Education and the State Administration of Foreign Experts Affairs of China.

**Conflicts of Interest:** The authors declare no conflict of interest

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
