# Peer review of "Effect of Ca Content on the Mechanical Properties and Corrosion Behaviors of Extruded Mg–7Li–3Al Alloys"

_metals, doi:10.3390/met9111212_

Round 1

Reviewer 1 Report

The authors have provided solid experimental results and analysis. After addressing the following comments, the paper should be ready for publication:

(1) Fig. 1&2: How did the authors distinguish alpha-Mg and beta-Li phase from the optical microscope images? did the authors run EDS measurement prior to that? if yes, those data should be included in this manuscript.

(2) If we look at the XRD pattern for Al2Ca, there is no significant increase in the peak intensity of for this compound with increasing Ca concentration. How did the authors confirmed that the amount of Al2Ca was indeed increased with increasing Ca content?

(3) Page 11: the authors mentioned :"AL2Ca are chemically more noble and more stable than alpha-Mg+beta-Li matrix". Please clarify this sentence.

(4) Fig. 13: The authors mentioned that this images were obtained after different corrosion times, but did not specify the corrosion times for Fig. 13 (a) to (f). This should be clarified as well.

(5) I could not find a clear explanation to why addition of 0.4Ca resulted in higher corrosion resistance than 0.8Ca or 1.2Ca. Furthermore, a clear correlation between the mechanical properties characterization and the corrosion resistance are not provided in this manuscript.

(6) Please double check and triple check the manuscript, because I found some type in the text.

(7) The authors used several abbreviations (such as TYS or Ecorr)but did not provide information about them. Please provide the full name for them.

Author Response

We would like to thank you for your constructive comments and suggestions, which have helped us improve the manuscript. According to your suggestions, we have revised our manuscript, which are highlighted in red in the manuscript. Moreover, an individual response to each comment/suggestion is detailed below:Please see the attachment

Reviewer 2 Report

I have no significant comments since the authors have nicely described the effect ot Ca addition fir the MgLi7Al3 alloy. sample preparation, analysis, and characterization of the physical properties are well executed. 

Author Response

We would like to thank you for your constructive comments and suggestions, which have helped us improve the manuscript.Thank you for your comment.

Reviewer 3 Report

This work describes the effect of the Ca additive on the mechanical, microstructural and corrosion properties of Mg-Li-Al alloys. The presented results are interesting and enrich the knowledge about metal alloys with anticorrosive agents. The whole work is read in an accessible way, the results are clearly presented, there is no problem with their analysis. In connection with the above, this work deserves to be published in the journal Metals, however it requires a few editorial corrections:

Introduction (line 25) – please explain term “3C” Results (lines 74,76) and rest part of manuscript – please standardize the recording units by the numbers Line 147 – Please correct caption Table 5 – explain which values? Table 6 (line 180) – please correct the table to be in the range between the page margins Lines 233-234 – please change the font

Author Response

Thank you for your comments. The paper has been revised carefully according to your suggestions, which are highlighted in red in the revised manuscript.
